# The First Pulmonary Hypertension Registry in the United Arab Emirates (UAEPH): Clinical Characteristics, Hemodynamic Parameters with Focus on Treatment and Outcomes for Patients with Group 1-PH

**DOI:** 10.3390/jcm12051996

**Published:** 2023-03-02

**Authors:** Khaled Saleh, Naureen Khan, Kelly Dougherty, Govinda Bodi, Miriam Michalickova, Samiuddin Mohammed, Theodora Kerenidi, Ziad Sadik, Jihad Mallat, Samar Farha, Hani Sabbour

**Affiliations:** 1Respiratory Institute, Cleveland Clinic Abu Dhabi, Abu Dhabi P.O. Box 112412, United Arab Emirates; 2St. George’s University, West Indies, Grenada; 3Temple University Hospital, Philadelphia, PA 19140, USA; 4Department of Pharmacy Services, Cleveland Clinic Abu Dhabi, Abu Dhabi P.O. Box 112412, United Arab Emirates; 5Critical Care Institute, Cleveland Clinic Abu Dhabi, Abu Dhabi P.O. Box 112412, United Arab Emirates; 6Cleveland Clinic Lerner College of Medicine, Case Western Reserve University, Cleveland, OH 44106, USA; 7Faculty of Medicine, Normandy University, UNICAEN, ED 497, 14032 Caen, France; 8Department of Pulmonary Medicine, Respiratory Institute, Cleveland Clinic, Ohio, OH 44195, USA; 9Heart and Vascular Institute, Cleveland Clinic Abu Dhabi, Abu Dhabi P.O. Box 112412, United Arab Emirates

**Keywords:** pulmonary hypertension, hemodynamic parameters, outcomes, idiopathic pulmonary hypertension, connective tissue disease, congenital heart disease, Riociguat, ambrisentan, selexipeg, sildenafil, iloprost, treprostenil, UAE

## Abstract

Background: The aim of this study is to present the first United Arab Emirates pulmonary hypertension registry of patients’ clinical characteristics, hemodynamic parameters and treatment outcomes. Method: This is a retrospective study describing all the adult patients who underwent a right heart catheterization for evaluation of pulmonary hypertension (PH) between January 2015 and December 2021 in a tertiary referral center in Abu Dhabi, United Arab Emirates. Results: A total of 164 consecutive patients were diagnosed with PH during the five years of the study. Eighty-three patients (50.6%) were World Symposium PH Group 1-PH; nineteen patients (11.6%) were Group 2-PH due to left heart disease; twenty-three patients (14.0%) were Group 3-PH due to chronic lung disease; thirty-four patients (20.7%) were Group 4-PH due to chronic thromboembolic lung disease, and five patients (3.0%) were Group 5-PH. Among Group 1-PH, twenty-five (30%) had idiopathic, twenty-seven (33%) had connective tissue disease, twenty-six (31%) had congenital heart disease, and five patients (6%) had porto-pulmonary hypertension. The median follow-up was 55.6 months. Most of the patients were started on dual then sequentially escalated to triple combination therapy. The 1-, 3- and 5-year cumulative probabilities of survival for Group 1-PH were 86% (95% CI, 75–92%), 69% (95% CI, 54–80%) and 69% (95% CI, 54–80%). Conclusions: This is the first registry of Group 1-PH from a single tertiary referral center in the UAE. Our cohort was younger with a higher percentage of patients with congenital heart disease compared to cohorts from Western countries but similar to registries from other Asian countries. Mortality is comparable to other major registries. Adopting the new guideline recommendations and improving the availability and adherence to medications are likely to play a significant role in improving outcomes in the future.

## 1. Introduction

Pulmonary hypertension (PH) is a pathophysiological disorder that may involve multiple clinical conditions and may be associated with a variety of cardiovascular and respiratory diseases [1].

It is characterized by an elevation in pulmonary artery pressure because of many different disease processes affecting the pulmonary vasculature. The World Symposium on pulmonary hypertension classifies PH into five major groups: Group 1-PH; Group 2-PH due to left heart disease; Group 3-PH due to lung diseases and/or hypoxia; Group 4-chronic thromboembolic PH (CTEPH) and other pulmonary artery obstructions; and Group 5-PH with unclear and/or multifactorial mechanisms [2]. Group 1-PH is a rare disorder of pulmonary vasculature that results in the progressive narrowing of the pulmonary arteries and increased vascular resistance that ultimately leads to right heart failure and death [3]. In 1987, the FRENCH registry was the first national cohort that reported on Group 1-PH patients’ clinical characteristics and outcomes [4]. Since then, many registries have been published in North America [5,6,7] and Europe, [8,9,10,11,12] indicating the progressive changes in clinical characteristics and outcomes due to improvements in diagnosis and treatment. Registries’ experiences from national or single centers were also published in Asian countries, namely Japan [13], China [14] and South Korea [15], followed by countries in Western Asia including Saudi Araba [16,17,18] and Turkey [19]. Here, we present the first PH registry from a tertiary PH referral center in the United Arab Emirates.

## 2. Materials and Methods

This is a single-center, retrospective study of consecutive patients over the age of 14 at Cleveland Clinic Abu Dhabi who were known or suspected of having PH between January 2015 and December 2021. Patients with a previously confirmed diagnosis are defined as the prevalent cases and the newly diagnosed ones as the incidence cases [20]. All patients had echocardiography, N-terminal pro B-type natriuretic peptide (NT-proBNP) and a 6-min walk test (MWT) before having right heart catheterization (RHC). A ventilation-perfusion (VQ) scan and/or high-resolution chest CT were performed on patients to rule out chronic thromboembolic pulmonary hypertension (CTEPH) or parenchymal lung disease, respectively. All patients had complete pulmonary function tests (PFT), abdominal ultrasound, hepatic virology screen and liver function tests, and serologies for connective tissue diseases or hyper-coagulable states when suspected. A diagnostic RHC was required for the diagnosis of PH and to be included in the study. Referred patients who had an RHC conducted in an outside hospital and not repeated in our hospital had their pulmonary artery wedge pressure (PAWP) confirmed by evaluating the wave form tracing for accuracy. If a wedge could not be measured, the left ventricular end-diastolic pressure was directly measured. No fluid challenges were performed to evaluate for heart failure with preserved ejection fraction.

The study protocol was approved by the Research Ethics Committee at Cleveland Clinic Abu Dhabi.

The clinical characteristics including age, sex, body mass index (BMI) and presence of comorbid diseases, such as diabetes (DM), hypertension (HTN), hyperlipidemia (HLD), coronary artery disease (CAD), chronic kidney disease (CKD) defined by glomerular filtration rate (GFR) less than 60% and hematologic diseases, were collected from the medical record. The updated classification of PAH was based on definitions outlined in the 2018 6th World Symposium of Pulmonary Arterial Hypertension [20]. PAH was defined by mean pulmonary artery pressure (mPAP) ≥25 mmHg, wedge pressure (WP) ≤15 mmHg and Pulmonary Vascular Resistance (PVR) ≥3 woods unit. Clinical data for PAH risk stratification were collected as recommended by the 6th World symposium including WHO functional class, NT-proBNP and six-minute walk test (MWT) in meters [21]. The echocardiographic findings, such as right atrial enlargement (mild, moderate and severe), presence or absence of pericardial effusion and transannular planar systolic excursion (TAPSE), were collected on the first and last echocardiography over the follow-up period. A positive nitric oxide (NO) vasoreactivity response was defined as a drop in mPAP of more than 10 mmHg to reach an absolute value of less than 40 mmHg with an increase or no change in cardiac output (CO). We used inhaled NO, 40 parts per million for 5–10 min, before hemodynamic measurements were repeated.

Pulmonary vasodilator medications were reported on the first and last visits. The medications used were phosphodiesterase-5 inhibitors (sildenafil or tadalafil), a soluble guanylate cyclase inducer (riociguat) and an endothelin receptor antagonists (ambrisentan or Macitentan). Prostacyclin analogs Selexipag, nebulized iloprost or subcutaneous/intravenous prostacyclin Treprostinil were used as the third medication when clinically indicated. In addition, supportive therapy anticoagulants (for CTEPH, and coexistent AFib/flutter), diuretics and oxygen therapy were used in accordance with ESC/ERCs 2015 and WSPH 2018 guidelines [22].

All patients in Group 2-PH were classified as having heart failure with preserved ejection fraction (HFpEF), based on echocardiographic findings, and right heart catheterization after optimization of their heart function with standard therapy by consultant cardiologists [23]. Patients in Group 3-PH were diagnosed based on significant interstitial lung disease on a chest CT scan involving at least 20% of lung parenchyma and/or FVC less than 60% predicted, which would explain the pulmonary hypertension related to chronic lung disease [24]. Patients in Group 4-PH had a V/Q scan and pulmonary angiogram as the gold standard for diagnosis of CTEPH [25].

### Statistical Analysis

All data were expressed as mean ± SD or as median (25–75%, interquartile range, (IQR)), as appropriate. The normality of data distribution was assessed using the Kolmogorov–Smirnov test. Comparisons of values between different groups or subgroups were performed by one-way ANOVA test or Kruskal–Walli’s test, as appropriate. Pairwise comparisons between different study times for continuous variables were assessed using paired Student’s t-test or Wilcoxon test, as appropriate. Comparisons of categorical data between the different groups or subgroups were performed using the Chi2 or Fisher’s exact tests, as appropriate. Pairwise comparisons between different study times for categorical variables were assessed using McNemar’s test. The cumulative probability of survival was analyzed using the Kaplan–Meier method. Patients who did not experience the event (death) during the study period were censored at the time of last seen. Differences between survival curves were assessed using the log-rank test.

Statistical analysis was performed using STATA 14.2 (StataCorp LP, College Station, TX, USA). A value of *p* <0.05 was considered statistically significant. All reported *p* values are 2-sided.

## 3. Results

During the study period, a total of 164 patients were diagnosed with pulmonary hypertension. The clinical characteristics and hemodynamic findings for Group 1–4 PH are shown in Table 1. Five patients diagnosed as Group 5-PH were not included due to the small number of patients in this group.

The majority of patients were referred to our center. The cohort consisted of 68% prevalent cases and 32% incident cases with the overall majority being female (76%).

Of the 83 patients in Group 1-PH, 25 (30%) had idiopathic, 27 (33%) had connective tissue disease (CTD) and 26 (31%) had congenital heart disease (CHD) (Table 2). Five patients (6%) had porto-pulmonary hypertension, two of whom had liver transplantations in our hospital with significant improvement in their hemodynamics and remained on the same vasodilator therapy. They were not included in the analysis due to the small number of patients in this subgroup. None of the patients had HIV or drug-induced PAH. Moreover, none of the patients were tested for schistosomiasis due to low clinical suspicion, as the UAE is not considered an endemic area.

Among the idiopathic subgroup, two patients had positive NO vasodilatory test (out of the nine tested) and had responded well to high-dose calcium channel blockers for a few years. One patient was suspected of having pulmonary veno-occlusive disease (PVOD) due to flash pulmonary edema after starting vasodilator therapy and died a few weeks after the diagnosis.

Among the Group 1-PH CTD subgroup, sixteen patients (59.2%) had scleroderma, five (18.5%) had systemic lupus erythematosus, three (7.4%) had Sjorgren Syndrome, one (3.7%) had rheumatoid arthritis and two (3.7%) had mixed connective tissue disease. In the CHD subgroup, there were nine unrepaired atrial septal defects (ASD), three repaired ASD, six unrepaired ventricular septal defects (VSD), one repaired VSD, four unrepaired patent ductus arteriosus and three patients who were post-Fontan surgery.

The patients in the CTD subgroup were the oldest; meanwhile, those in the CHD subgroup were the youngest and had the lowest BMI (Table 2). There were no differences regarding the WHO classifications and the six-MWTs among the subgroups. The CHD group had the highest mPAP and CI by the modified Fick method. The CTD had the lowest PVR (both by modified Fick and thermodilution). There were no differences in the mixed venous saturations.

In terms of medication treatment during the first visit, the majority of referred patients were on monotherapy of phosphodiesterase 5 inhibitor (PDE5i). Most of the incident cases were started on dual therapy following confirmation by RHC. At the follow-up visit, the majority of patients were escalated from dual therapy to triple therapy **(**Figure 1).

Considering the treatment effects in the three subgroups over the follow-up period with a median of 55.6 months [IQ, 15–107], only the CHD group had statistically significant improvement in the 6-MWT and RVSP on the echocardiography. There were no differences in terms of RA size enlargement changes, RV dysfunction changes, and NTproBNP in the three subgroups (Table 3).

Follow-up invasive hemodynamics assessment over the treatment period was not performed routinely as per standard guidelines [22]. Only about one-quarter of the patients required repeat RHC because of clinical deterioration. Therefore, the follow-up hemodynamic data were not included in the analysis.

We used the Simplified French model as a non-invasive risk stratification tool for follow-up visits, utilizing three low-risk criteria: WHO FC 1/2, 6-MWT > 440 m and NT-proBNP < 300 ng/L [26,27].

At follow-up, 20% of the patients in the idiopathic subgroup had a decrease in at least one low-risk criterion, 24% remained unchanged and 56% improved in at least one low-risk criterion compared to 26%, 29% and 55% for CTD and 23%, 38% and 39% for the CHD subgroups, respectively.

The one-, three- and five-year cumulative probabilities of survival for Group 1-PH and for the subgroups are shown in Table 4 and Figure 2 and Figure 3.

## 4. Discussion

In this study, we present the clinical characteristics, hemodynamics and comorbidities of patients suspected of having PH at a tertiary referral center in the United Arab Emirates. Group 1-PH constituted about 50% of the patients followed by Group 4-PH (21%). Group 2-PH had the most common predefined comorbid conditions, namely DM, hyperlipidemia, CAD and atrial fibrillation, which are known risk factors for heart failure with preserved ejection (HFpEF). None of the patients in Group 1-PH had evidence of diastolic dysfunction or more than two risk factors for HFpEF. This is important to prevent the misclassification of patients with HFpEF in Group 1-PH, considering that the treatment effects and outcomes of patients with these risk factors are quite different than those with a low risk for left ventricular dysfunction [28,29].

Our hospital is the only referral center for PH in the UAE with access to all citizens as well as the uninsured, supported by a funded mandate for adults with life-threatening cardiovascular conditions. Therefore, we believe that most, if not all, patients diagnosed with idiopathic PH are being followed in our center. Based on the reported census of the UAE population in 2021 of about 10 million [30], the estimated prevalence of idiopathic pulmonary hypertension is 8 to 9 cases per million individuals, which is lower than the reported estimated prevalence of 15–50 per million [31]. This could be related to under or delayed diagnosis. Fifty-four percent of the patients were Emirati nationals. This reflects the unique UAE demographics with a high number of expatriates living in the country. Few patients in the cohort were of Caucasian and South Asian descent; the rest were from the Middle Eastern and North African regions.

The median age of the Group 1-PH patients was 41 years, which is likely due to the higher prevalence of younger CHD patients in this group. This is closer to the reported mean age of 35 years in the Saudi registry (SAUDIPH) [16] but quite different than the western registries of reported average age in the early 50s [12]. The percentages of the 3 subgroups in our study are somewhat different from the reported percentage ranges of 30–50% idiopathic, 15–30% CTD and 10–23% CHD subgroups reported in the western registries [5,31]. The relative proportion is similar to non-western countries where CHD is the more common subtype, i.e., 43% and 41% in the Chinese and the Saudi registries, respectively [14,16].

Most of our patients (62%) were in WHO Class III/IV failure at baseline indicating late referral, particularly in patients with idiopathic PH where 71% were in Class III/IV on the first visit. Over the follow-up period, the CTD subgroup had the most improvement in WHO classification but did not meet the statistical significance (*p* = 0.07). This is different than what has been reported in terms of better clinical improvement with vasodilator therapy in idiopathic compared to patients with the CTD subgroup [32]. This also occurred despite more patients in the idiopathic subgroup being advanced to triple therapy than in the CTD subgroup (Table 3). It is conceivable that the idiopathic subgroup patients were diagnosed later in the disease process than their CTD counterparts due to earlier recognition of the PH in the latter group by the referring rheumatologists who were aware of the risk of PH in this patient population.

At the follow-up visit, a considerable number of patients in the three subgroups were switched from sildenafil or tadalafil to riociguat. This was based on an open-label study showing that riociguat might be used as an escalation of therapy rather than an escalation to oral prostanoid therapy, which became available to our center in late 2019 [33].

Most of our patients had significant improvement in at least one low-risk criterion, which is shown in idiopathic PH as the indicator for survival probability [26]. However, further improvement could have been achieved with more aggressive therapy. This can be explained in part due to the limited resources for some patients to receive IV or subcutaneous prostacyclin analogue or due to interruptions of medication supply or noncompliance at some point in the treatment period (particularly during the COVID 19 pandemic), which is reported to negatively impact the outcome [34].

The access to medications has significantly improved since mid-2019 after the UAE Federal Ministry of Health approved an innovative patient-support program through a collaboration between the pharma industry and charitable organizations. The objective of the program is to cover these high-cost medications to patients that are not covered by their health insurance providers. This will likely improve our patients’ clinical outcomes in the future.

Our one-, three- and five-year cumulative probabilities of survival for Group 1-PH and its three subgroups (Table 3) are comparable to the survival analysis of Group 1-PH’s cumulative and respective three subgroups by the largest single-center cohort of 685 patients in Germany, as reported in 2017 [12].

However, our results were appreciably lower than the survival probabilities in the SAUDIPH registry, reporting the cumulative survival probabilities 96, 89 and 75% [16]. This can be explained by the somewhat younger population (mean age 31 vs. 41), the lower percentage of patients with connective tissue disease (11% vs. 32%) and the lower percentage of patients in WHO Class III/IV (55 vs. 62%) in the SAUDIPH compared to the UAEPH counterpart.

Since there has been a significant improvement in survivability due to upfront combination therapy and the addition of a triple oral agent, we compared our data for the idiopathic subgroup patients to the most recent registry in the US, enrolling patients since 2015. Their reported one-, three- and five-year probabilities of survival for the idiopathic subgroup were 92% (95% CI, 6–10%), 84% (95% CI, 13–19%) and 79% (95% CI, 17–25%), which is somewhat comparable to our data [35].

Our study limitations include the retrospective nature and the relatively small number of patients, considering that the registry is only seven years old. We also acknowledge the under-utilization of repeat right heart catheterization which can be helpful in risk stratification, particularly in patients who did not improve in low-risk criteria. There was limited data on medication adherence, limitations in escalating therapy due to cost and how changes in follow-ups during the COVID-19 pandemic period could have affected the outcome. Many of the patients in the cohort were expatriates from different parts of the world, which limits the generalizability of the study.

## 5. Conclusions

This is the first registry of Group 1-PH from a single tertiary referral center in the UAE. Our cohort had younger patients and a higher percentage of CHD patients compared to Western countries’ cohorts but was similar to registries reported from other Asian countries. Mortality is comparable to other major registries. Adopting the recently updated guideline recommendations and improving the availability of and adherence to medications (particularly parenteral prostanoids) are likely to play a significant role in improving outcomes in the future.

## Figures and Tables

**Figure 1 jcm-12-01996-f001:**
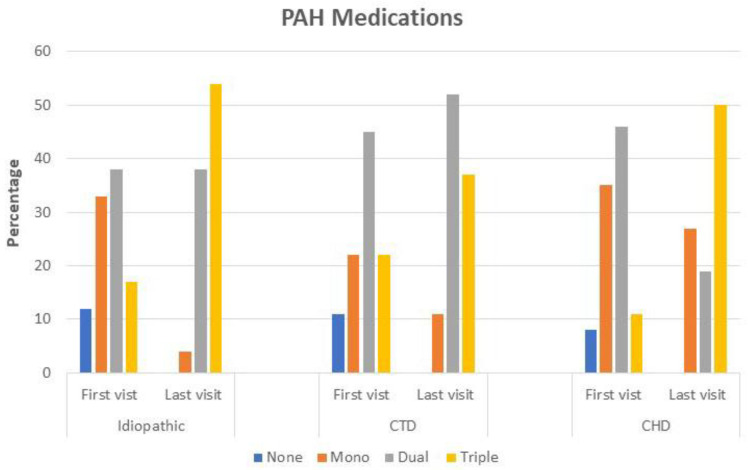
None, mono, dual and triple vasodilator therapies for Group 1-PH subgroups at initial and last visit. CTD, connective tissue disease; CHD, congenital heart disease.

**Figure 2 jcm-12-01996-f002:**
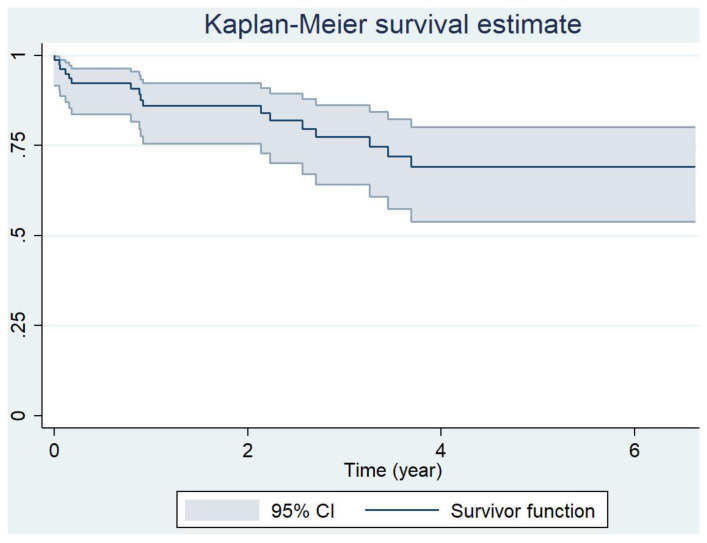
Survival for the overall group 1-PH cumulative.

**Figure 3 jcm-12-01996-f003:**
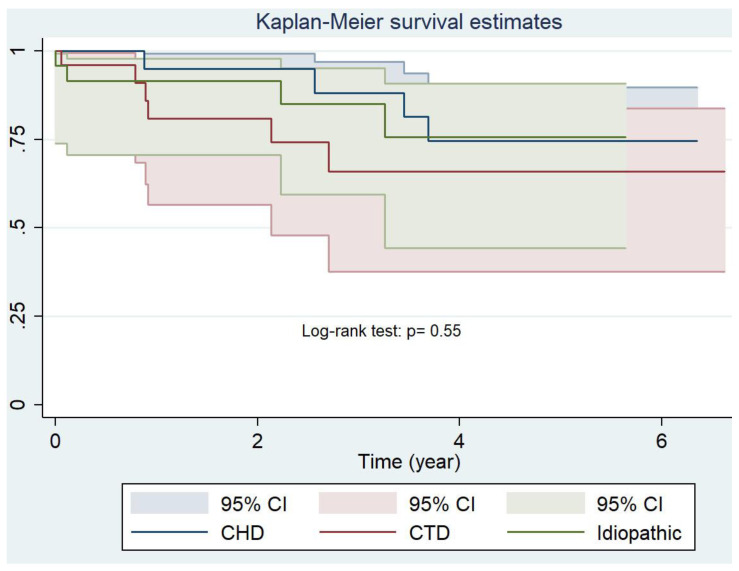
Survival for the subgroups 1-PH cumulative. CHD, congenital heart disease; CTD, connective tissue disease-related pulmonary hypertension.

**Table 1 jcm-12-01996-t001:** Baseline demographic, clinical characteristics and hemodynamics of all Group 1–4 PH patients.

Variable	All (n = 159)	Group 1 (n = 83)	Group 2 (n = 19)	Group 3 (n = 23)	Group 4 (n = 34)	*p*-Value
Age, year	47 [35–64]	41 [32–54]	70 [59–76]	59 [53–69]	42 [33–60]	<0.001
Female, n (%)	122 (76)	65 (78)	16 (80)	18 (78)	23 (68)	0.65
Prevalence, n (%)	118 (74)	68 (82)	15 (75)	12 (52)	23 (68)	0.03
BMI, kg/m^2^	27.8 [22.6–31.8]	27.2 [21.9–20.1]	29.9 [25.2–36.6]	27.7 [21.4–33.5]	28.4 [23.8–34]	0.19
Follow-up, month	55.6 [15.2–107.0]	67.7 [16.2–135.3]	55.4 [20.2–87.7]	43.4 [8.2–68.8]	48.3 [18.2–119.4]	0.57
Essential hypertension, n (%)	66 (42)	25 (30.5)	18 (90)	12 (54.6)	11 (32.4)	<0.001
Diabetes, n (%)	37 (23.4)	11 (13.4)	12 (60)	6 (26.1)	8 (24.2)	<0.001
Hyperlipidemia, n (%)	(22.7)	(17.1)	(47.1)	(27.3)	(21.2)	0.068
Coronary artery disease, n (%)	17 (10.7)	6 (7.2)	4 (20)	4 (17.4)	3 (9.1)	0.22
Atrial fibrillation, n (%)	15 (9.4)	4 (4.8)	9 (45)	2 (8.7)	0 (0)	<0.001
Pulmonary embolism, n (%)	32 (20)	4 (4.8)	0 (0)	2 (8.7)	26 (76.5)	<0.001
Chronic kidney disease, n (%)	17 (10.6)	3 (3.6)	9 (45)	2(8.7)	3 (8.8)	<0.001
Hematologic ds	31 (19.4)	13 (16.7)	3 (15)	3 (13)	12 (35.3)	0.098
Baseline 6-MWT, m	280 [172–393]	320 [201–407]	216 [97–306]	110 [75–384]	297 [238–366]	0.03
Baseline NT-proBNP, pg/mL	484 [146–1726]	315 [96–1178]	1164 [392–2808]	402 [210–2889]	668 [148–1810]	0.05
TAPSE, cm	1.8 [1.5–2.1]	1.8 [1.6–2.1]	2 [1.7–2.4]	1.6 [1.4–2.2]	1.6 [1.5–2.1]	0.46
WHO III/IV, n (%)	103 (65)	51 (62)	12 (60)	17 (77)	24 (68)	0.56
mRAP, mmHg	8 [4–12]	8 [3–13]	11 [9–13]	6 [1–9]	9 [4–12]	0.009
mPAP, mmHg	45 [35–55]	48 [42–62]	42 [33–49]	38 [29–43]	48 [32–53]	0.005
PAWP, mmHg	10 [7–16]	10 [6–16]	18 [12–19]	10 [5–14]	11 [8–12]	0.008
CI (Fick method), L/min/m^2^	2.5 [2–3]	2.7 [2–3.1]	2.1 [1.9–2.5]	2.7 [2.3–3.2]	2.1 [2–2.5]	0.03
PVR (Fick method), mmHg/L/min	590 (344–880)	779 (376–932)	441 [295.8–628.7]	452 [333–742]	660 [225–992]	0.06
CI (thermodilution method), L/min/m^2^	2.6 [2.0–3.6]	2.5 [2.0–3.9]	2.0 [1.9–3.1]	3.0 [2.3–3.6]	2.0 [1.8–3.2]	0.42
PVR (thermodilution method), mmHg/L/min	640 [350–834]	702 [334–945]	622 (338–1003)	398 [361–611]	975 (444–6258]	0.4
SVO_2_, %	64 [56–72]	66 [60–73]	62 [49–72]	66 [56–70]	57 [50–64]	0.06
LVEF, %	57 [57–67]	60 [58–67]	60 [55–67]	60 [60–65]	60 [57–65]	0.9
RSVP, mmHg	67 [52–86]	71 [52–91]	61 [42–78]	57 [53–81]	66 [54–85]	0.4
FVC, %	59 [43–75]	64 [48–77]	52 [42–76]	43 [31–58]	69 [56–81]	0.006
DLCO, %	48 [32–63]	54 [40–65]	60 [32–82]	21 [19–36]	53 [41–72]	0.006

BMI, body mass index; 6-MWT, six-minute walk test; NT-proBNP, N-terminal pro-brain natriuretic peptide; TAPSE, tricuspid annular plane systolic excursion; WHO, world health organization; mRAP, mean right atrial pressure; mPAP, mean pulmonary arterial pressure; PAWP, pulmonary arterial wedge pressure; CI, cardiac index; PVR, pulmonary vascular resistance; SVO_2_, venous oxygen saturation; LVEF, left ventricle ejection fraction; RSVP, right systolic ventricle pressure; FEV1, forced expiratory volume during the first second; DLCO, diffusing capacity for carbon monoxide. Data are expressed as median and [interquartile 25–75] or count and percentage (%).

**Table 2 jcm-12-01996-t002:** Baseline clinical and hemodynamic characteristics of Group 1-PH subgroups.

Variables	Idiopathic (n = 25)	CTD (n = 27)	CHD (n = 26)	* p * -Value
Age, year	39 [31–53]	47 [41–66]	33 [26–41]	<0.001
Female, n (%)	19 (79)	25 (93)	18 (69)	0.09
BMI, kg/m^2^	28.2 [24.8–31.6]	28.5 [22–31.5]	23.4 [18.3–26.5]	0.03
Prevalence, n (%)	19 (79)	25 (93)	21 (81)	0.30
WHO III/IV, n (%)	17 (71)	17 (65)	12 (46)	0.17
Baseline 6-MWT, m	280 [182.9–378]	332 [201–218]	274 [198–374]	0.77
Baseline NT-proBNP, pg/mL	1161 [304–2250]	324 [95–585]	182 [79–490]	0.006
mRAP, mmHg	10 [2–15]	7 [3–7]	8 [3–14]	0.60
mPAP, mmHg	55 [46–63]	43 [23–52]	67 [44–73]	0.004
PAWP, mmHg	10 [6–16]	9 [6–13]	8 [6–17]	0.80
CI (Fick method), L/min/m^2^	2.3 [1.9–2.7]	2.7 [2.0–3.0]	3.2 [2.3–3.4]	0.03
PVR (Fick method), mmHg/L/min	842 [540–1201]	531 [212–908]	837 [745–931]	0.07
CI (thermodilution method), L/min/m^2^	2.3 [1.8–2.9]	2.4 [2.0–4.3]	2.8 [2.5–4.1]	0.15
PVR (thermodilution method), mmHg/L/min	735 [658–975]	406 [131–918]	733 [581–793]	0.04
SVO_2_, %	66 [57–72]	69 [57–73]	69 [61–74]	0.50
TAPSE, cm	1.8 [1.6–2.0]	1.7 [1.5–2.5]	1.8 [1.6–2.0]	1.00
LVEF, %	61 [55.5–67]	60 [59–67]	60 [56–64]	0.70
RSVP, mmHg	81 [56–97]	54 [41–79]	86 [70–97]	0.004

CTD; connective tissue disease; CHD, congenital heart disease; 6-MWT, six-minute walk test; NT-proBNP, N-terminal pro-brain natriuretic peptide; TAPSE, tricuspid annular plane systolic excursion; WHO, world health organization; mRAP, mean right atrial pressure; mPAP, mean pulmonary arterial pressure; PAWP, pulmonary arterial wedge pressure; CI, cardiac index; PVR, pulmonary vascular resistance; SVO2, venous oxygen saturation; LVEF, left ventricle ejection fraction; RSVP, right systolic ventricle pressure. Data are expressed as median and [interquartile 25–75] or count and percentage (%).

**Table 3 jcm-12-01996-t003:** Changes in clinical, echocardiographic parameters and PH medications at diagnosis and last follow-up visit for Group 1-PH according to subgroups.

Variables	Idiopathic (n= 25)			CTD (n = 27)			CHD (n = 26)		
	Baseline	Last Visit	*p*-Value	Baseline	Last Visit	*p*-Value	Baseline	Last Visit	*p*-Value
6-MWT, m	280 [183–378]	389 [171–469]	0.11	332 [201–418]	358 [229–454]	0.77	274 [198–374]	392 [255–471]	0.01
NT-proBNP, pg/mL	1161 [304–2250]	341 [109–1783]	0.57	324 [94–585]	202 [90–2836]	0.32	183 [79–490]	225 [118–604]	0.17
WHO III/IV, n (%)	17 (71)	14 (58)	0.55	17 (65)	11 (41)	0.07	12 (46)	10 (38)	0.73
RSVP, mmHg	81 [56–97]	71 [46–93]	0.32	54 [42–79]	51 [40–80]	0.77	86 [70–97]	85 [66–106]	0.03
Reduced RV function, n (%)	16 (73)	14 (67)	1	11 (48)	7 (39)	1	13 (54)	11 (61)	0.69
Enlarged RA, n (%)	20 (87)	16 (80)	0.68	10 (42)	5 (33)	1	14 (70)	11 (79)	1
Medications									
Sildenafil/Tadalafil, n (%)	14 (58)	12 (50)	0.69	13 (48)	8 (30)	0.12	16 (62)	9 (35)	0.04
Riociguat, n (%)	1 (4)	16 (67)	<0.001	6 (22)	12 (44)	0.07	0 (0)	14 (54)	<0.001
Endothelin RA, n (%)	12 (50)	20 (83)	0.02	19 (70)	24 (89)	0.12	21 (81)	24 (92)	0.37
Iloprost, n (%)	3 (12)	2 (8)	1	4 (15)	4 (15)	1	5 (19)	5 (19)	1
Selexipag, n (%)	0 (0)	4 (16)	0.12	0 (0)	5 (18)	0.06	0 (0)	5 (19)	0.06
Treprostinil, n (%)	8 (33)	9 (41)	1	3 (11)	7 (26)	0.25	0 (0)	3 (12)	0.25
None	3 (12)	0 (0)	0.62	3 (11)	0 (0)	0.25	2 (8)	0 (0)	1
Single	8 (33)	1 (4)	0.04	6 (22)	3 (11)	0.45	9 (35)	7 (27)	0.62
Dual	9 (38)	9 (38)	1	12 (45)	14 (52)	0.45	12 (46)	5 (19)	0.04
Triple	4 (17)	13 (54)	0.02	6 (22)	10 (37)	0.22	3 (11)	13 (50)	0.002

CTD; connective tissue disease; CHD, congenital heart disease; 6-MWT, six-minute walk test; NT-proBNP, N-terminal pro-brain natriuretic peptide; WHO, world health organization; RV, right ventricle; RA, right atrium; RVSP, right ventricle systolic pressure. Data are expressed as median and [interquartile 25–75] or count and percentage (%).

**Table 4 jcm-12-01996-t004:** Survival probability (%) of cumulative Group 1-PH and three subgroups.

Probability of Survival	Cumulative Group 1 (n =83)	Idiopathic (n = 25)	CTD (n = 27)	CHD (n = 26)
1st year, % (95% CI)	86 (75–92)	90 (66–97)	82 (60–93)	95 (71–99)
3rd year, % (95% CI)	69 (54–80)	73(41–90)	66 (38–84)	75 (45–90)
5th year, % (95% CI)	69 (54–80)	73 (41–90)	66 (38–84)	75 (45–90)

CTD; connective tissue disease; CHD, congenital heart disease; CI, confidence interval.

## Data Availability

The data presented in the study are available on request from the corresponding authors. The data are not publicly available du to ethical restrictions.

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
