# Peer review of "The First Pulmonary Hypertension Registry in the United Arab Emirates (UAEPH): Clinical Characteristics, Hemodynamic Parameters with Focus on Treatment and Outcomes for Patients with Group 1-PH"

_jcm, 2023, doi:10.3390/jcm12051996_

Round 1

Reviewer 1 Report

The paper provides results of the first PH registry in UAEPH describing in detail patients’ clinical characteristics, hemodynamic parameters, and treatment outcome. Ethic committee approved the study. The manuscript is clear, relevant to topic and well structured. Data is presented in tables and figures, is clear, but need some minor corrections and verification (details below).  Conclusions follow the results obtained in the study. References are related to the topic and up-to date. 

Suggestions for correction:

-line 82/82 the sentence is not clear, what means "outside", section "methods" states this is single center study, please clarify if results were obtain elsewhere.

-in Table 1 -results of PAWP for group 1, 75% IRQ reaches 16mmHg, could you comment on this finding.

-in Table 2 the results of mPAP for CTD , 25% IRQ is 23 mmHg, could you comment on this finding.

- in Table 2 the results of PVR for CTD , 25% IRQ is 212 mmHg/L/min, could you comment on this finding.

-in Figure 1 please correct word to "idiopathic".

-in Figure 1 consider removing "0" from picture, it makes it difficult to read 

-Table 3, please verify data/statistics results, number of patients is small, be sure the results are correct.

-The only group in which NT-proBNP increased is CHD group (not significantly), despite improvement in 6MWT and RVSP, could you comment on that in results

-Line 229 - please correct abbreviation - "HLP".

Author Response

#Reviewer #1

Comments and Suggestions for Authors

Reviewer: The paper provides results of the first PH registry in UAEPH describing in detail patients’ clinical characteristics, hemodynamic parameters, and treatment outcome. Ethic committee approved the study. The manuscript is clear, relevant to topic and well structured. Data is presented in tables and figures, is clear, but need some minor corrections and verification (details below). Conclusions follow the results obtained in the study. References are related to the topic and up-to date. 

Response: We thank the reviewer for this supportive comment.

Reviewer: line 82/82 the sentence is not clear, what means "outside", section "methods" states this is single center study, please clarify if results were obtain elsewhere.

Response: We thank the reviewer for this comment. This reflects a wide variation in practices outside the center of excellence. If patients were transferred from other facilities with incomplete workup or non-diagnostic RHC was performed, then it would be repeated with precise delineation of hemodynamics. However, if RHC was performed in another hospital, the PAWP value was confirmed by checking the waveform tracing. We have now added the following sentence to the manuscript: “Referred patients who had RHC done in an outside hospital and not repeated in our hospital had their pulmonary artery wedge pressure (PAWP) confirmed by evaluating the waveform tracing for accuracy.” We hope that we have addressed the reviewer’s issue appropriately.

Reviewer: in Table 1 -results of PAWP for group 1, 75% IRQ reaches 16mmHg, could you comment on this finding.

Response: We thank the reviewer for this comment. All right heart cath tracings, especially PAWP, were directly analyzed by two experts in invasive hemodynamics, and the respiratory variation was accounted for. Correlation with echocardiography and left atrial pressures and diastolic parameters was also performed to ensure the definition of pre-capillary pulmonary hypertension is precisely met. Advanced cases were presenting late with RV dysfunction resulting in LA compression and elevated PAWP in some patients. These patients had high PVR and low risk for heart failure with preserved ejection fraction and were well scrutinized to prevent misclassification. We hope that we have addressed the reviewer’s issue appropriately.

Reviewer: in Table 2 the results of mPAP for CTD, 25% IRQ is 23 mmHg, could you comment on this finding.

Response: We thank the reviewer for this comment. Patients who were referred to us early by a rheumatologist had lower PAP, and some patients were already on either PED5i or ERA for suspected PAH or Raynaud. Wash-out of these medications, when possible, was performed 3-4 days prior to the diagnostic RHC. Other patients (expatriates) were treated in their respective countries and established their follow-up care at our center and had a follow-up right heart cath reflecting prior treatment. We hope that we have addressed the reviewer’s issue appropriately.           

Reviewer: in Table 2 the results of PVR for CTD, 25% IRQ is 212 mmHg/L/min, could you comment on this finding. 

Response: We thank the reviewer for this comment. Same as above, both mPAP and PVR were lower than expected. We hope that we have addressed the reviewer’s issue appropriately.           

Reviewer: in Figure 1 please correct word to "idiopathic".

Response: We apologize for the typo. We have corrected it in Figure 1.         

Reviewer: in Figure 1 consider removing "0" from picture, it makes it difficult to read 

Response: Another option is to put “none” instead of “0”. However, this carries same meaning.  We believe putting a number (0) is more consistent with the data presented than a word (none). We hope that we have addressed the reviewer’s issue appropriately.

Reviewer: Table 3, please verify data/statistics results, number of patients is small, be sure the results are correct.

Response: We thank the reviewer for this comment. We agree the number of patients is considered small. We have the data reviewed by two physicians and a statistician to confirm the accuracy. We found some errors regarding the p-value for the medication comparisons that we corrected in the manuscript. However, these do not change the main findings of the study. We thank the reviewer for her/his thorough reading of the manuscript. We hope that we have addressed the reviewer’s issue appropriately.

Reviewer: The only group in which NT-proBNP increased is CHD group (not significantly), despite improvement in 6MWT and RVSP, could you comment on that in results.

Response: We thank the reviewer for this comment. Due to CHD-associated co-morbidities, there are confounding variables such as AF and compliance that caused nonsignificant variation in the NtPro BNP as well. NTpro BNP is low in CHD, both at baseline and follow-up, so the changes are neither statistically nor clinically significant. Both fall in the low clinical risk criteria. We hope that we have addressed the reviewer’s issue appropriately.

Reviewer: Line 229 - please correct abbreviation - "HLP". 

Response: We thank the reviewer for this comment. We have now corrected “HLP” to “hyperlipidemia”.        

We greatly appreciate your time and input to improve our manuscript and to illustrate some clarifications.

Reviewer 2 Report

First description of PH in UAE

Worthwhile endeavour and useful contribution to worldwide registry data

Well written manuscript

The Kaplan Meier survival plots could be edited for improved aesthetics

Author Response

#Reviewer #2

Comments and Suggestions for Authors

Reviewer: First description of PH in UAE. Worthwhile endeavour and useful contribution to worldwide registry data. Well written manuscript.

Response: We appreciate your input and encouragement.

Reviewer: The Kaplan Meier survival plots could be edited for improved aesthetics.

Response: We thank the reviewer for this comment. We have removed the 95% CI from Figure 3 to make it more aesthetic. We hope that we have addressed the reviewer’s issue appropriately.

We greatly appreciate your time and input to improve our manuscript and to illustrate some clarifications.